# FDTD Simulations of Sweat Ducts and Hair at 0.45 THz

Zoltan Vilagosh [1,2,*], Negin Foroughimehr [1,2], Alireza Lajevardipour [1,2] and Andrew W. Wood [1,2]

1   School of Health Sciences, Swinburne University of Technology Melbourne, Hawthorn, VIC 3122, Australia
2   Australian Centre for Electromagnetic Bioeffects Research, Hawthorn, VIC 3122, Australia
*   Correspondence: zvilagosh@swin.edu.au

**Abstract:** Advances in Terahertz frequency electromagnetic radiation (THz) production technologies have produced an increasing interest in exploring possible applications. New applications will inevitably lead to increased incidental interaction of humans with THz radiation. Given that the wavelength of THz radiation is in the same order of magnitude as the dimensions of skin structures such as hair and sweat ducts, the possibility of interaction among these structures is of interest. The interaction was studied utilizing Finite Difference Time Domain (FDTD) simulations using a far-field excitation of 0.45 THz. No antenna-like effects were detected. Regions of increased specific absorption rate (SAR) due to reactive near-field effects with both the hair and sweat ducts were found in the order of 0.01–0.05 mm and 0.001–0.002 mm, respectively. Simulations using unwound sweat ducts yielded the same penetration pattern as the helical structure, indicating that the helical structure has no impact on the propagation of THz radiation in skin.

**Keywords:** sweat duct; THz; hair shaft; antenna

## 1. Introduction

The Terahertz (THz) frequency electromagnetic radiation band spans 0.1 to $10.0 \times 10^{12}$ Hz, corresponding to wavelengths in empty space of 3 mm to 0.03 mm. Whilst THz radiation is emitted from any black body radiation source, there are no significant natural sources that would impact on the human skin, and the Earth's atmosphere absorbs any solar THz radiation. THz radiation has found applications in security screening of parcels and personnel [1], and manufacturing processes such as non-destructive quality control and quantitative analysis of chemical mixtures [2–4].

THz has a higher frequency compared to the current mobile data and communication bands, giving it potential for greater data-carrying capacity. This has led to proposals for telecommunication applications using frequencies up to 0.30 THz [5,6]. There are no data on the effects of long-term exposure of humans to THz radiation [7,8].

The radiation from the THz band is highly absorbed by water, with an absorption coefficient ($\alpha$) in the order of 80 to 3500 cm$^{-1}$ [9,10]. The water content of most body tissues is 70% to 75%, and given the high absorption coefficient of water in THz, the effective tissue penetration of the radiation at body temperature is in the order of 0.01–0.3 mm. It follows that most of the absorption of THz will be in the epidermis, outer layers of the dermis, the cornea and the tympanic membrane. The effects of THz radiation on normal human skin, the cornea and the human tympanic membrane have been explored previously by the group [11–13].

The dimensions of the human hair and sweat ducts is the same order of magnitude as the wavelength of 0.3–1.5 THz radiation. This leads to questions regarding the possible production of resonant or antenna-like effects leading to increased radiation exposure and also confounding methods of image production.

The skin is divided into thin and thick skin types, with the chief differentiating factor being the dimensions of the outer stratum corneum (SC) layer, which is 0.01 to 0.03 mm deep in thin skin [14] and 0.15 to 0.50 mm [15] in thick skin. Thick skin is found on the

palm of the hand and the sole of the foot. The significance of the SC is its relatively low hydration. Thick skin SC has 15–40% hydration, whilst deeper skin layers contain 70–72% free water [16], keeping with most other soft tissues. The hydration of the outer SC is dependent on the level of sweating as well as environmental factors such as humidity and temperature. Given the THz absorption by water, the hydration level in the dead SC significantly reduces the amount radiation reaching the deeper living tissue for a given incident power flux density (PD).

Hair is only found on the thin skin and is associated with oil-producing sebaceous glands. The human hair shaft has a complex morphology [17]. It is between 0.025 and 0.05 mm in radius. Hair thickness has individual and ethnic variation as well as variation with levels of hydration [18]. Water uptake can change in the diameter of the hair by as much as 12%.

The sweat glands that are found in both thin and thick skin are mostly of the eccrine type. The apocrine sweat glands are found in areas such as the axillae (arm pits), eyelids and nostrils. The human sweat duct becomes a helix in the epidermis, with a right-handed preference. The radius of the helix is 0.045–0.050 mm and the pitch angle is ~12° [15]. This yields a rise of 0.02 mm per turn. It flows that the average number of turns in the 0.05 to 0.10 mm Stratum Spinosum (SS) is 2–5, while the number of turns in the SC in thin skin is 0.5 to 1.5, and there can be up to 25 turns in the SC of thick skin.

The helical nature of the sweat ducts has sparked discussion on whether these structures act as an antenna at frequencies in the 0.3 to 0.5 THz range [19]. The concept was extended to the region of 0.3 to 0.45 THz in thick skin by Hayut et al. [20] and Tripathi et al. [15].

The helical antenna equation predicts that the optimal response is in the range $3\lambda/4 \leq 2\pi r \leq 4\lambda/3$, where $\lambda$ is the wavelength in the medium that surrounds the antenna and $r$ is the radius of the helix. With a radius of 0.045–0.05 mm, the optimal wavelength in the medium is 0.22 to 0.40 mm.

The matter is complicated by the change in the refractive index ($n$) (which changes $\lambda$) as the properties of the medium that surrounds the antenna change with tissue type, hydration and the frequency of the incident radiation. For example, the $n$ for human nail (a proxy for a dry SC) is 1.72 at 0.45 THz and 1.69 at 1.0 THz [21]; on the other hand, the $n$ for non-keratinized colonic lining (proxy for the deeper skin tissues) is 2.10 at 0.45 THz and 2.00 at 1.0 THz [22].

The optimal incident wavelength for the antenna to function thus changes with the tissue involved. In thick skin, where most of the sweat duct is in the SC, ($n$~1.7) provides an incident, in air, optimal "antenna" frequency of 0.44 to 0.79 THz. On the other hand, in the living epidermal layers of the sweat ducts that reside in the Stratum Spinosum (SS) ($n$~2.10), making the incident, in air, optimal "antenna" frequency of 0.35 to 0.64 THz.

The radiative output of any antenna is divided into the near and far fields. Far-field radiation is understood to be achieved only after the individual regions of any antenna have interacted and have produced a cohesive pattern. The far field is assumed to start at a distance (R), using the empirical equation of $R > 2D^2/\lambda$, with D being the maximum linear dimension of the antenna and $\lambda$ being the wavelength of the radiation in the medium the antenna is radiating into. In the case where D < $\lambda$, R is better described by $R > (2D^2/\lambda) + \lambda$ [23].

At 0.45 THz, within the SS, with the $n$ of 2.10 (Table 1), there is a $\lambda$ of $\approx$ 0.32 mm, and the absorption coefficient ($\alpha$) for the SS is 103 cm$^{-1}$. If the width of the hair is taken as the value for D (0.1 mm, thus < $\lambda$), then the start of the far field, R, is at about 0.38 mm. An $\alpha$ of 103 cm$^{-1}$ reduces the radiated signal intensity to 0.02 of the original at a distance of 0.38 mm. It follows that the near-field antenna effects are the only ones likely to have any impact, as the absorption of THz is too high to allow any antenna to radiate significantly into the far field. It is worth noting that that much of antenna theory relies on empirical equations derived from highly conductive antennas radiating into air or free space; thus, the behavior of juxtaposed, very lossy dielectric biological tissues may not follow "antenna theory".

**Table 1.** Simulations at 0.45 THz (λ = 0.667 mm).

| Model Type | Short Hair Thin Skin | Minimal Hair Thin Skin | Long Hair Thin Skin | Sweat Ducts Thin Skin | Sweat Duct Thick Skin |
|---|---|---|---|---|---|
| Problem Space (Yee cells) | $841 \times 552$ $\times 454$ | $173 \times 169$ $\times 139$ | $717 \times 475$ $\times 536$ | $841 \times 552$ $\times 454$ | $875 \times 587$ $\times 375$ |
| Maximum cell dimension | $\lambda/82$ | $\lambda/95$ | $\lambda/82$ | $\lambda/82$ | $\lambda/26$ |
| Minimum cell dimension | $\lambda/2392$ | $\lambda/167$ | $\lambda/1499$ | $\lambda/2392$ | $\lambda/335$ |
| Step size (fs) | 0.499 | 2.917 | 0.581 | 0.499 | 1.322 |
| Number of Timesteps | 20,000 | 10,000 | 30,000 | 20,000 | 30,000 |
| Simulated time (ps) | 14.1 | 29.2 | 17.4 | 14.1 | 23.9 |
| Oscillations/Simulated time | 6.4 | 13.2 | 7.8 | 6.3 | 9.8 |

For the sweat ducts, assuming four coils in thin skin, giving a D = 0.28 mm, R becomes > 2.08 mm, and the far-field equation gives a proportion of the radiated signal reaching the far field of ~$5 \times 10^{-10}$ of the original. The absorption in the very lossy SS results in practically no radiation reaching the far field; only the near-field aspects of any antenna-like activity for the sweat dust helices need to be considered in thin skin.

Sweat ducts in thick skin are longer (0.35 mm in the model) and are embedded mainly in the dead SC. Since the SC has a lower *n*, there is a longer λ for any given frequency. The SC also has a lower absorption coefficient. Taking an example of 15% hydration (*n* = 1.76, *α* = 54), R ≈ 1.0 mm, which still gives a proportion of the signal reaching the far field of 0.0045.

Given that the near-field antenna effects are the only ones likely to have any impact, the near-field pattern of any antenna is dependent on the interaction of small components of the antenna with neighboring components. Kuster and Balzano [24] noted that lossy biological tissues at frequencies above 300 MHz display anomalous SAR patterns that can be best explained by magnetic field (H-field)-induced currents and that "SAR values are not always consistent, and some results and differences are even qualitatively not satisfactorily explainable in physical terms". Christ et al. [25], in a study spanning 60 MHz to 6.0 GHz, produced standing waves in the tissues and found that under some conditions and at distances of approximately λ/40, "reactive E-field components lead to high local absorption in the skin". Cell membrane permeability was noted using high-intensity 18 GHz radiation [26]. These studies are helpful, but they were conducted at frequencies which were 200 to 1000 times lower than the THz frequencies under consideration in this paper.

The standards for exposure limits to THz radiation set by the International Commission on Non-Ionizing Radiation Protection (ICNIRP) fall into two categories. The standards for exposure of greater than 0.30 THz are contained in the guidelines for laser radiation of wavelengths between 180 nm and 1000 μm [27]. The standards are expressed as the maximum incident power flux density (PD); for durations greater than 10 s, the standard for maximum exposure is 1 kWm$^{-2}$, and for short pulses of less than 100 ns, the standard is expressed as a total energy exposure of 100 Jm$^{-2}$.

Whilst the PD is used for exposure standards, the detailed assessment interaction of skin appendages with THz radiation also requires the estimation of the specific absorption rate (SAR) surrounding the appendages of the skin. Given the dimensions of layers such as the SB, a resolution in the order of 0.01 mm is desirable.

The main interest in the dosimetry study of skin appendages is in the 0.30 to 0.70 THz range (λ of 1.0 to 0.43 mm in empty space), as this range has been used for exploring communication links, imaging skin morphology [28,29]. This is also the range of the reported THz sweat duct studies. A 0.01 mm resolution translates to λ/100 at 0.30 THz and λ/43 at 0.70 THz in free space. When the *n* of ~2.0 for the tissues is considered, the resolution becomes ~λ/50 and ~λ/22 at 0.30 THz and 0.70 THz, respectively.

THz production and spectroscopy methods are improving; however, the existing THz detectors can only discern a resolution in the order of 0.05 mm [30]. This limits the

direct exploration of THz dosimetry, exposure patterns and the potential for diagnostic imaging. No experimental information exists on the distribution of radiation absorption nor is there any detail of the SAR within or surrounding the hair or sweat ducts in the 0.30 to 0.70 THz range.

Given the current technical limitations to resolution, computational modeling becomes an attractive method for the preliminary exploration of interaction of the skin appendages on a sub-wavelength scale, which can reveal areas of anomalous absorption as well as describe interactions that may confound image production.

Computational modeling extends knowledge beyond the capabilities of current technology, generating ideas for future applications as THz technology improves. Imaging THz is a qualitatively different method to the other imaging techniques on offer or in development and can offer perspectives unachievable by other means [31]. The presence of hair and sweat ducts needs to be studied in detail, as their presence may confound the imaging by changing or distorting the received signal.

As with all computational modeling, computational phantom THz skin appendage studies are limited by the precision of the inputs. The rendering of faithful anatomical models is important, but in the case of skin appendage/THz studies, the use of reliable dielectric properties of tissues is paramount. The dielectric properties of skin and its appendages are incompletely understood in the THz frequency band; however, it is possible to make realistic assumptions using available data and supplementing the dataset from tissue proxies and the use of mixing formulae.

## 2. Materials and Methods

The computational method employed was the Finite Difference Time Domain (FDTD) type. The method was first defined by Yee [32] and expanded by [33]. The FDTD method is described by Sullivan [34]. An FDTD solver XFdtd Bio-Pro (version 7.6.0.5.r48456, Remcom, State College, PA, USA) was used for the bulk of the design of the anatomical models and implementation of the simulations.

The frequency of 0.45 THz (15 waves cm$^{-1}$, 0.667 mm wavelength, in free space) was selected for the simulations. This choice relied on a review of 32 papers by Vilagosh [35], which revealed that most data dealing with the interaction of skin and THz were available in the range 0.2 to 1.2 THz. The 0.45 THz frequency is the approximate geometric mean of the most useful range of 0.30–0.70 THz and is within the range of the "sweat duct antenna" theoretical calculations.

The incident excitation in all cases was a liner polarized continuous far field source of 1.0 Vm$^{-1}$. The simulation becomes contaminated with unwanted lateral reflections, which places a practical limit on the duration of the simulations. The time can only be extended by increasing the size of the entire model. It was found that due to the poor penetration of 0.45 THz radiation beyond the Stratum Spinosum, the computation times used were adequate for the simulation of hair and sweat ducts in both thin and thick skin.

To adequately explore the thin skin hair and sweat ducts within the computational limitations, the simulation was performed using variable geometry. The minimum resolution, problem space, number of time steps, time-step duration, and total simulation time are outlined in Table 1. Since the refractive index of skin tissues is in the order of 2, the indicative λ within skin is in the order of 0.33 mm at 0.45 THz (λ = 0.667 mm in air).

To maximize resolution, the thin skin models' cell sizes ranged from λ/82 for less important areas to λ/2382 for the detail around sweat ducts and rete ridges. To accommodate the larger dimensions of the SC in the thick skin, the thick skin model was limited in design to a maximum of λ/26 and a minimum of λ/335. Since the refractive index of skin tissues is in the order of 1.7 to 2.1, the λ at 0.45 THz within skin was about 0.32–0.39 mm. In practice, as little as 0.45 THz radiation penetrated to skin layers past the dead layer of the SC in the thick skin simulations, and the grainier method was adequate.

The rendering of the anatomical detail of the skin and appendages was limited by the capacity of the computer hardware; the random-access memory available for the models

was limited the maximum model size to 25 gigabytes. Separate models for thin skin with hair, thin skin with sweat ducts and thick skin with sweat ducts were employed. All models had an anatomical representation of the skin layers, including rete ridges, an irregular surface and a "fingerprint pattern" in the case of thick skin.

The interaction of THz radiation with hair was studied with three models, each of 0.05 mm radius: a basic keratin rod (Figure 1A), a composite hair (Figure 1B) and an extended hair model (Figure 1C). In the basic model, the hair shaft is represented as a homogenous cylinder with a beveled upper edge.

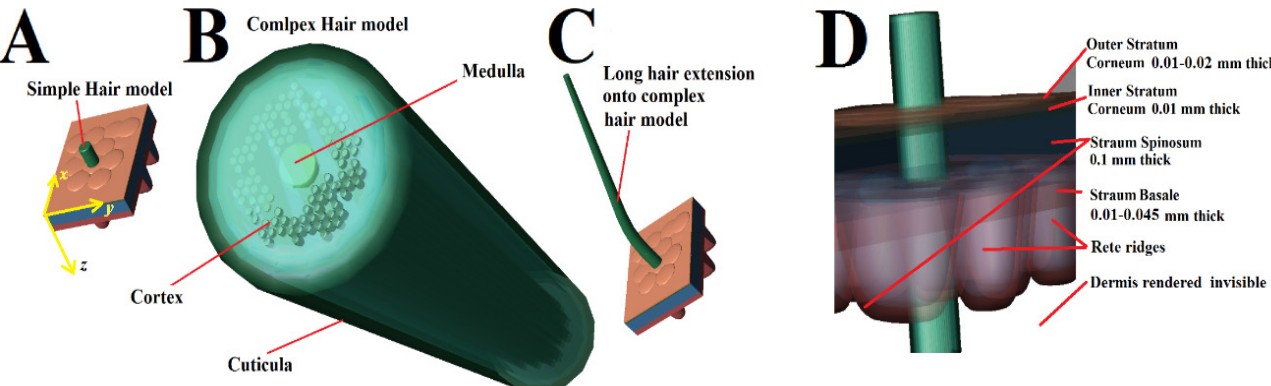

**Figure 1.** Dermis is rendered invisible in the illustrations. (**A**) The basic hair model, with a solid hair rod, 0.05 mm radius, and total length 0.6 mm, extending 0.15 mm above the skin. (**B**) Complex hair model, with external dimensions and placement as in (**A**) but with internal structure of a cortex containing microfibrils and a medulla. (**C**) An extension adding a 1.7 mm bent, conical, tapering shaft to the complex model. (**D**) Detail of the skin layers, thin skin.

The complex model has an internal structure within the hair shaft. The models were embedded in thin skin as there is no hair in thick skin, protruding 0.15 mm above the surface. The extended hair model was used to explore the behavior of an uncut hair, with an addition of a 1.7 mm bent, conical, tapering extension placed above the complex model, giving a total above skin hair height of 1.85 mm. The extended model thus required an increase in the problem space of 1.7 mm in the z direction, which necessitated a reduction in the computational resolution.

The model was embedded in material with dielectric properties equivalent to the Dermis to prevent interference from lateral excitation, giving a total dimension of $x = 1.8$ mm, $y = 1.0$ mm, $z = 0.6$ mm. The thickness of the skin layers is presented in Figure 1D. A series of oblate spheroids 0.01 mm in height were embedded in the outer SC layer to represent an irregular, rough surface. In the thin skin, the outer SC layer was set at a hydration level of 15%, and the inner SC layer was set at a hydration level of 40%. The penetration of 0.45 THz radiation to the level of the sebaceous glands at 1.0 to 2.0 mm depth is very limited, and these were not represented. All hair models have representation of the skin layers as outlined in Figure 1D. The hair model was subjected to varying the angle of incident excitation from $\theta i = 0°$ (orthogonal to the skin) to $\theta i = 30°$ and $60°$.

The behavior of sweat ducts was studied with both the thin skin and thick skin models (Figure 2A–D). The dimensions and basic anatomy of the thin skin model described in Figure 2D, without the hair, were used for the sweat duct simulations in thin skin. The thick skin model is described in Figure 2B–D. The thick skin model had a fingerprint pattern on the surface and 3 further layers of SC, which could be independently set for hydrations. For the sweat duct simulations, the outer layers were set at hydrations of 15% for the fingerprint pattern and the upper layer, which was followed by 23% and 30% for the subsequent layers. A further layer, representing the Stratum Lucidum and Stratum Granulosum, was set at a hydration of 40%. The dermis in the thick skin was modeled 2.0 mm in thickness to monitor

the possible radiation pattern of the helical sweat duct. The dimensions of the thick skin model were $x$ = 3.55 mm, $y$ = 2.40 mm, $z$ = 2.30 mm.

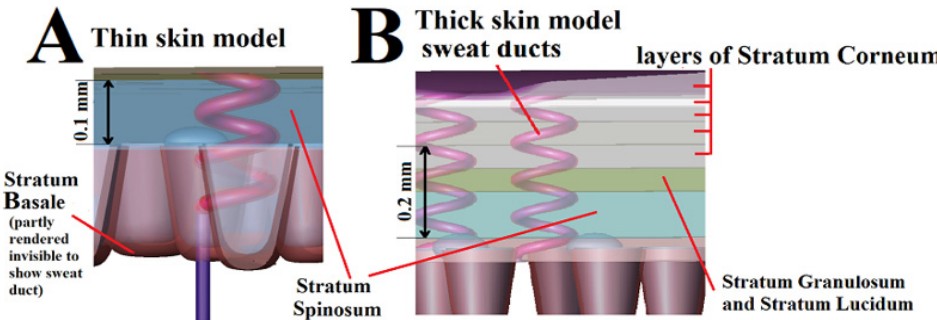

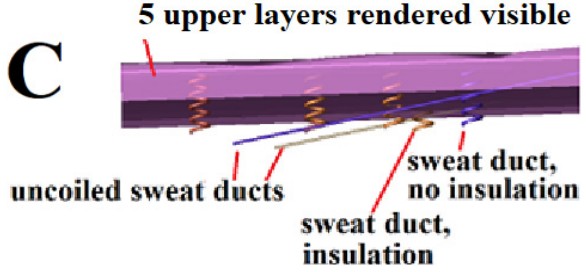

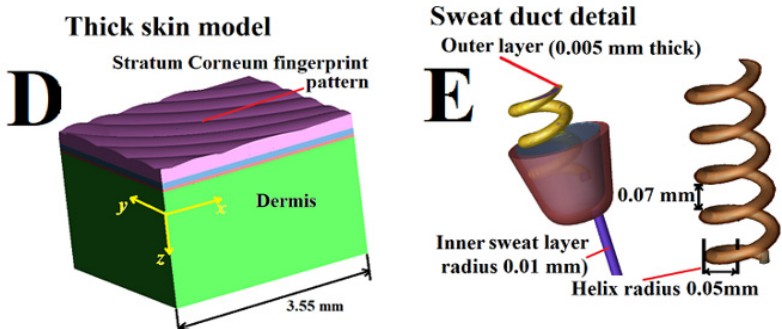

**Figure 2.** (**A**) The sweat duct within the XFdtd thin skin model, with 3 helical turns, chiefly within the stratum spinosum, positioned within the rete ridges to match anatomy. (**B**) Detail of the thick skin model, with the layers of the Stratum Corneum (gray), Stratum Granulosum and Strum Lucidum (green), Stratum Spinosum (blue), Stratum Basale and the rete ridges. (**C**) Multiple sweat ducts to be simulated concurrently each with distinct properties. The "uncoiled" sweat ducts are illustrated. (**D**) Overview of the thick skin model with fingerprint pattern. (**E**) Sweat duct detail: an independently variable outer coat of 0.005 mm thickness and an inner duct radius of 0.01 mm.

The sweat within the ducts was modeled with three conductivities. The first takes into account only the dielectric properties of the physiological sweat itself. The value is derived from an assumption that the sweat described by Braconnier et al. [36] is equivalent to 50 mmol/L NaCl solution with the THz values extrapolated from Jepsen et al. [37]. Since there is minimal variation in the imaginary ($\varepsilon''$) part of the complex permittivity between NaCl solutions, Jepsen et al. [37] report molar concentrations of 0 to 5.1 M having $\varepsilon''$ values tightly grouped in the 4.0–4.4 range at 0.45 THz; any individual variations of sweat concertation would have little effect on the underlying assumption of a $\varepsilon''$ of 4.2, giving a bulk physiological sweat conductivity ($\sigma$) value of 105 Sm$^{-1}$.

In addition, two assumptions reflect the upper and lower bounds (1000 Sm$^{-1}$ and 10,000 Sm$^{-1}$) of the published proposals that the sweat in the ducts has a much higher AC

conductivity due to local duct effects. This proposal was first made by Feldman et al. [38] and subsequently explored by Hayut et al. [20] and others. The justification of the proposed higher values for sweat conductivity used by Feldman can be found in [19].

To test the proposal by Feldman et al. [19], that the helical nature of the sweat duct produces an antenna effect in the skin, the sweat ducts were modeled as an anatomically correct sweat duct and also as "uncoiled" straight sweat ducts that were the same dimension and maintained the same pitch as the helical sweat duct but did not have a helical structure. The sweat duct in the thin skin was embedded in the basic model described in Figure 1D (without the hair) to produce the model shown in Figure 2A. Multiple sweat ducts were embedded in the thick skin model along with "uncoiled" ducts as with the thin skin model (Figure 2B,C). The dimensions of the thick skin model are shown in Figure 2D.

The anatomically correct sweat duct shown in Figure 2E could be filled with either "physiological" ($\sigma$ = 105 Sm$^{-1}$) sweat or the claimed high conductivity ($\sigma$ = 1001 Sm$^{-1}$) or very high conductivity ($\sigma$ = 10,014 Sm$^{-1}$) sweat. Four "uncoiled" sweat ducts were studied: two parallel with the $x$ axis and two aligned with the $y$ axis.

The incident radiation used was polarized; the "uncoiled" sweat ducts would be lined up either in the direction of the electric field or at right angles to it. One of each "uncoiled" pair was specified as having been filled with "physiological" sweat (105 Sm$^{-1}$) and the other was filled with very high conductivity (10,014 Sm$^{-1}$) sweat. To test the possibility that the outer rim of the sweat duct may act as an insulator or as a region of increased conductivity, the sweat ducts were designed to be able to independently change the dielectric properties of the outer 0.005 mm of the duct.

The dielectric values for the skin components, hair components and sweat, expressed as the real ($\varepsilon'$) and imaginary ($\varepsilon''$) parts of the complex permittivity, electrical conductivity ($\sigma$, Sm$^{-1}$), absorption coefficient ($\alpha$, cm$^{-1}$), refractive index (n), tissue density (m, kg m$^{-3}$) and heat capacity (c, J kg$^{-1}$ K$^{-1}$), are set out in Table 2. These parameters were used to calculate the SAR, PD and associated initial temperature rise.

**Table 2.** Dielectric Properties of Skin Components at 0.45 THz.

| | $\varepsilon'$ | $\varepsilon''$ | Electrical Conductivity $\sigma$ Sm$^{-1}$ | Absorption Coefficient $\alpha$ cm$^{-1}$ | Refractive Index $n$ | Tissue Density $\rho$ kgm$^{-3}$ |
|---|---|---|---|---|---|---|
| Stratum Corneum, hydrated 15% | 3.0 | 1.0 | 25.0 | 53.7 | 1.76 | 1300 |
| Stratum Corneum, hydrated 23% | 3.2 | 1.3 | 32.5 | 67.2 | 1.82 | 1260 |
| Stratum Corneum, hydrated 30% | 3.4 | 1.5 | 37.6 | 74.9 | 1.89 | 1230 |
| Stratum Corneum, hydrated 40% | 3.7 | 1.8 | 45.1 | 85.8 | 1.98 | 1200 |
| Stratum Spinosum | 4.1 | 2.3 | 57.6 | 103.0 | 2.10 | 1060 |
| Stratum Basale | 4.4 | 3.2 | 80.1 | 136.0 | 2.22 | 1060 |
| Dermis | 4.0 | 3.1 | 77.6 | 167.0 | 2.21 | 1080 |
| Hair Cuticle | 2.0 | 0.5 | 12.5 | 33.0 | 1.43 | 1300 |
| Hair Cortex | 2.2 | 0.45 | 11.3 | 28.4 | 1.49 | 900 |
| Hair Medulla | 2.6 | 0.35 | 8.8 | 20.4 | 1.62 | 1060 |
| Sweat, physiological | 5.0 | 4.2 | 105 | 165.0 | 2.40 | 1020 |
| Sweat, assumption of 1001 Sm | 5.0 | 40.0 | 1001 | 792.0 | 4.76 | 1020 |
| Sweat, assumption of 10,014 Sm | 5.0 | 400 | 10014 | 2650.0 | 14.2 | 1020 |

Any melanin content as a contributor to THz absorption was not considered. Setting the sweat conductivity at different levels has consequences for the complex permittivity, absorption coefficient, and refractive index. Using the "high conductivity" ($\sigma$ = 10,014 Sm$^{-1}$) sweat and maintaining the $\varepsilon'$ at 5.0 of "physiological" sweat results in an $\alpha$ of 2650 cm$^{-1}$ and n of 14.2 at 0.45 THz. These values are 20 times and 7 times higher, respectively, when compared to physiological sweat.

If, on the other hand, one maintains an α at 165 cm$^{-1}$ of "physiological" sweat, ε′ becomes 52,000, and there is an n of 228 in the high conductivity (σ = 10,014 Sm$^{-1}$) sweat. These values are 10,000 times and 100 times higher, respectively, when compared to physiological sweat. As noted, there is no direct evidence for the dielectric properties of sweat when they are contained in sweat ducts and thus no rational means of setting the other sweat parameters. The simulations were performed with the assumption of ε′ of "physiological" sweat for all the sweat types, which is in line with Feldman et al. [19]

Differences in the refractive index cause reflections and transmitted traveling or evanescent waves as the radiation moves from one medium to another. Evanescent waves are produced when the condition of total reflection is achieved. Evanescent waves at THz frequencies can give rise to complex frustrated total internal reflection phenomena [39]. Since the hair and sweat ducts are modeled as spherical structures, a changing continuum of the intensity of reflections and the transmitted traveling or evanescent waves is anticipated. As noted previously, computational modeling is inherently limited by the precision of the inputs. The THz dielectric parameters of water are well known, but the rest of the parameters rely on estimates gleaned from the references as indicated. It is difficult to give an overall, universal, error margin. An estimate for all values of +/− 15% would be reasonable.

Dielectric values are derived from data from Png et al. [39], Huang et al. [40], Jördens et al. [41], Sy et al. (2010) [42], Sim et al. (2013a, 2013b) [43,44], Guseva et al. [21], Yamaguchi et al. [45], Hernandez-Cardoso et al. [46,47], Hübers et al. [47] and Mizuno et al. [48] and the analysis of mixing formulae from Jördens et al. [40] and Ney and Abdulhalim [49]. The concertation of NaCl in sweat is derived from Braconnier et al. [36], and the complex permittivity is derived from Jepsen and Merbold [37]. The tissue densities are based on Hasgall et al. [50] and, for Stratum Corneum and hair cuticle, Dias et al. [51]

The SAR calculation was performed within the simulation using the equation:

$$\text{SAR} = \sigma \, E^{2/}\rho \tag{1}$$

where E, σ and ρ are the RMS electric field strength, electrical conductivity and tissue density, respectively. The equation used for the incident power density (PD) was:

$$\text{PD} = E^2/377 \tag{2}$$

where impedance air is equal to 377 Ω. The PD within the skin layers was calculated using the equation:

$$\text{PD} = n \, E^2/377 \tag{3}$$

where *n* is the refractive index of the relevant tissue, from [52]. Given that the SAR is calculated over the entire simulation, and PD varies with time with each oscillation, SAR is more accurate at estimating the total exposure of a region over time, and PD is more useful at illustrating the time-variant changes in the radiation exposure. In practice, both parameters were needed to understand the interaction of THz radiation with the skin appendages.

The sensor output of the SAR was a false color image of the absolute values. In addition, planar E-field sensors were placed in all models. These sensors yielded a false color; images of the absolute value of the E-field with a sampling interval was every 100 timesteps.

### 3. Results

*3.1. Hair Simulations*

Typical SAR cutplanes at $\theta_i = 0°$ (orthogonal to the skin) for the hair simulations are shown in Figure 3A–C. Vertical cutplanes are shown in Figure 3D–F. The dimensions of regions of increased SAR surrounding the hair are in the order of 0.01–0.05 mm. The SAR pattern in the simulations with a $\theta_i$ of 30° and 60° excitations demonstrated a greater increase on the side opposite the direction of excitation.

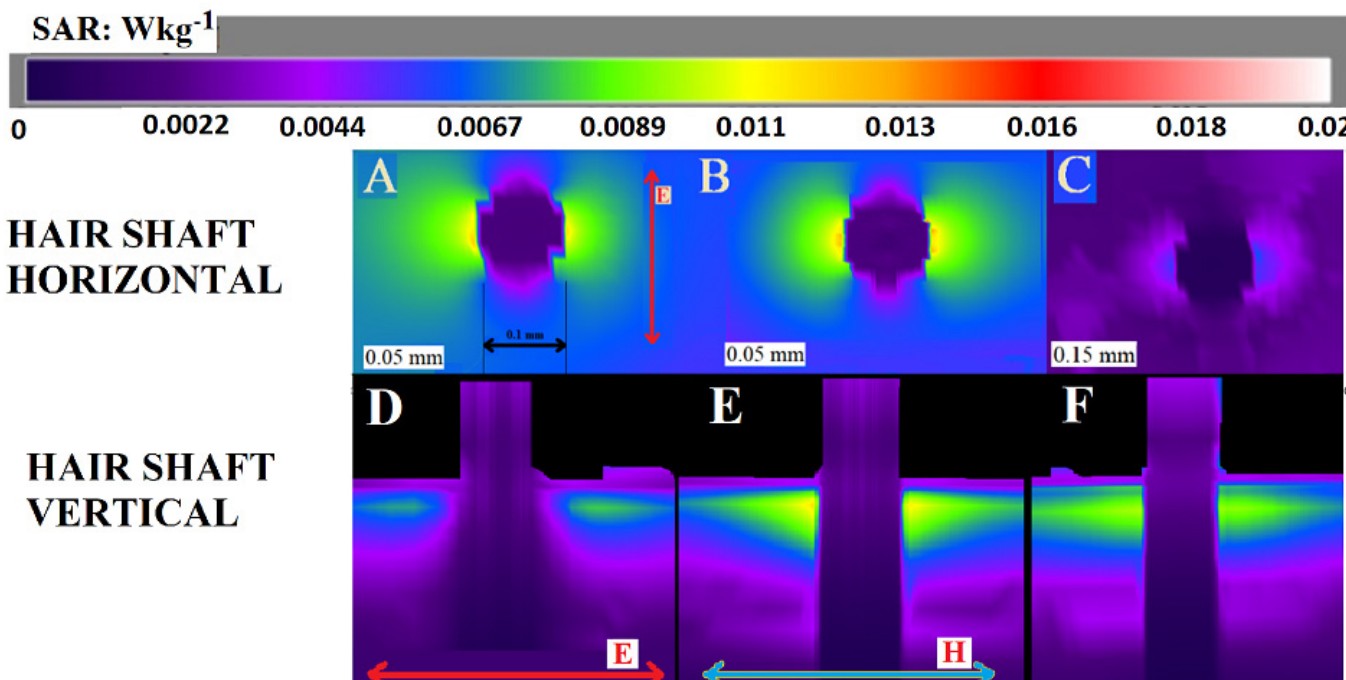

**Figure 3.** SAR distribution, $\theta_i = 0°$ (orthogonal to the skin), anatomically correct hair radius of 0.05 mm. (**A**) Horizontal cross-section, simple hair shaft, direction of E-field is shown (**B**) Complex hair shaft. (**C**) Simple hair at a depth of 0.15 mm (in the Stratum Basale). (**D**) Vertical cross-section, complex hair shaft, direction of E-field is shown. (**E**) Complex hair shaft, direction of H-field is shown (at right angles from (**D**)). (**F**) Simple hair, H-field as shown in (**E**).

The simulation results for the 30° and 60° angled excitations are shown in Figure 4. The SAR at a horizontal cutplane at 0.05 mm, complex hair, with $\theta i = 30°$, is demonstrated in Figure 4A. The excitation is from left, and the direction of the E-field is shown. A three-dimensional image, with a cutplane at 0.047 mm, is presented in Figure 4B. Figure 4C,D, demonstrate the time-domain images of the PD change in the lateral cutplane at $\theta_i = 30°$.

The time-domain PD change horizontal cutplane at 0.05 mm, at $\theta_i = 30°$ and 60°, is shown in Figure 4E–G and Figure 4H–J, respectively. To improve contrast, the PD in the values for the $\theta_i = 30°$ figures are presented over the range of 0–0.004 $Wm^{-2}$ and for $\theta_i = 60°$, the range is 0–0.002 $Wm^{-2}$.

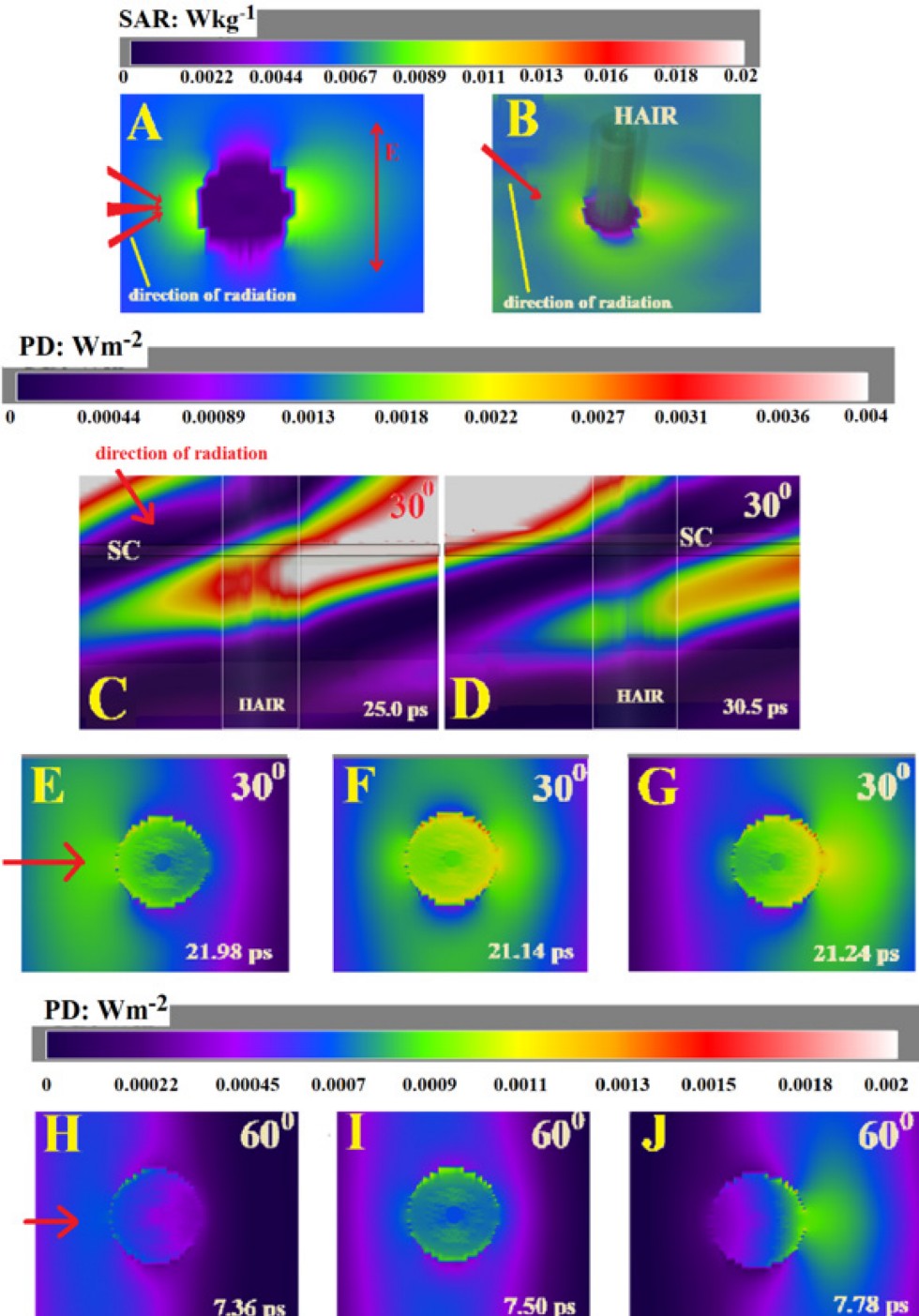

**Figure 4.** E-field direction as shown in (**A**), direction of the incoming radiation shown by the red arrows. (**A**) Horizontal cross-section, SAR distribution at 0.05 mm, $\theta i = 30°$. (**B**) A three-dimensional complex hair view. (**C**,**D**) Vertical cross-section, PD progression, $\theta_i = 30°$, excitation from upper left, complex hair. The position of the SC and the hair shaft are shown. (**E**–**G**) Horizontal section, PD progression, $\theta_i = 30°$, excitation from left. (**H**–**J**) Horizontal cross-section, PD progression, $\theta_i = 60°$, excitation from left. The frames demonstrate the asymmetrical distribution of PD and SAR surrounding the hair, which changes with the incident angle. The pattern is due to the interaction of reflection and the traveling and evanescent wave formation in the hair shaft.

*3.2. Sweat Duct Simulations*

The simulation results for the helical sweat ducts are presented for the thin skin in Figure 5 and for the thick skin in Figure 6.

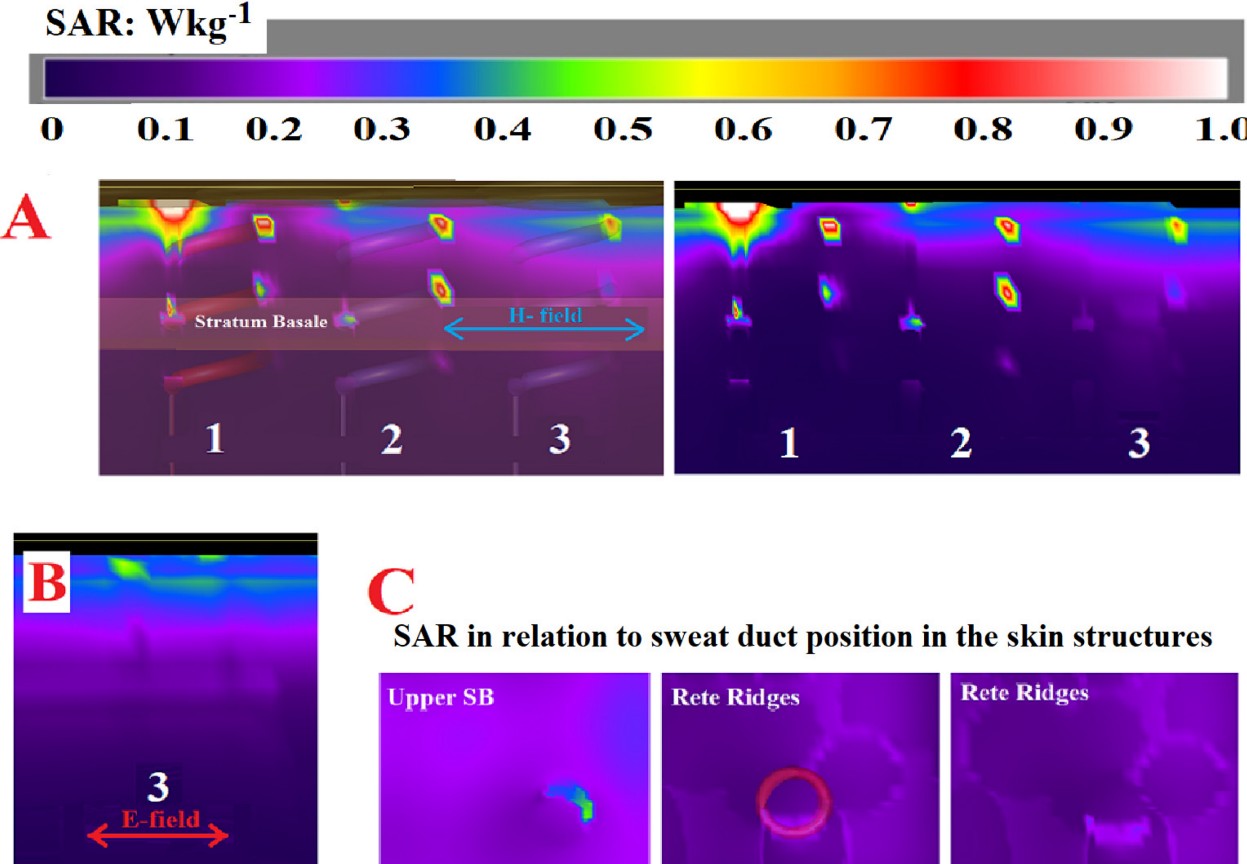

**Figure 5.** Simulations of SAR surrounding sweat ducts in thin skin. The duct (1) represents a sweat duct containing 1001 $Sm^{-1}$ sweat, duct (2) contains 10,014 $Sm^{-1}$ sweat and duct (3) is filled with "physiological" conductivity sweat of 105 $Sm^{-1}$. (**A**) Vertical cross-section, direction of the polarized H-field as shown (E-field is into the page) SAR. Left panel showing the general position of the sweat ducts and the Stratum Basale. There are minor differences in the distribution of SAR in the ducts containing "high electrical conductivity sweat" of 10,014 $Sm^{-1}$ and 1001 $Sm^{-1}$. The left edge of the duct containing the "physiological" sweat (105 $Sm^{-1}$) is barely discernible. (**B**) "Physiological" sweat containing a duct, vertical section, the E-field as shown, polarization at 90° to the direction in (**A**). (**C**) Horizontal cross-sections, "physiological" sweat containing duct, at 0.12 mm depth (the level of the Stratum Basale) and at 0.16 mm (deeper at the rete ridges).

In the SS of thin skin, some regions next to all the sweat ducts showed no increase in SAR above background levels, whilst in other regions, the SAR was 800% above background with the physiological sweat and 1000% above background for the sweat ducts containing the "high electrical conductivity" 1001 $Sm^{-1}$ and 10,014 $Sm^{-1}$ sweat. In the Stratum Basale, the simulations of "physiological" sweat containing ducts displayed areas of PD and SAR increase of 200% above the background penetration levels.

The Stratum Basale penetration using the "high electrical conductivity" sweat displayed PD and SAR regions of 800% for the background immediately adjacent to the ducts. Similarly, for the hair, the regions of increased SAR and PD were approximately the diameter of the sweat duct (0.02 mm), which would translate to less than one cell thickness.

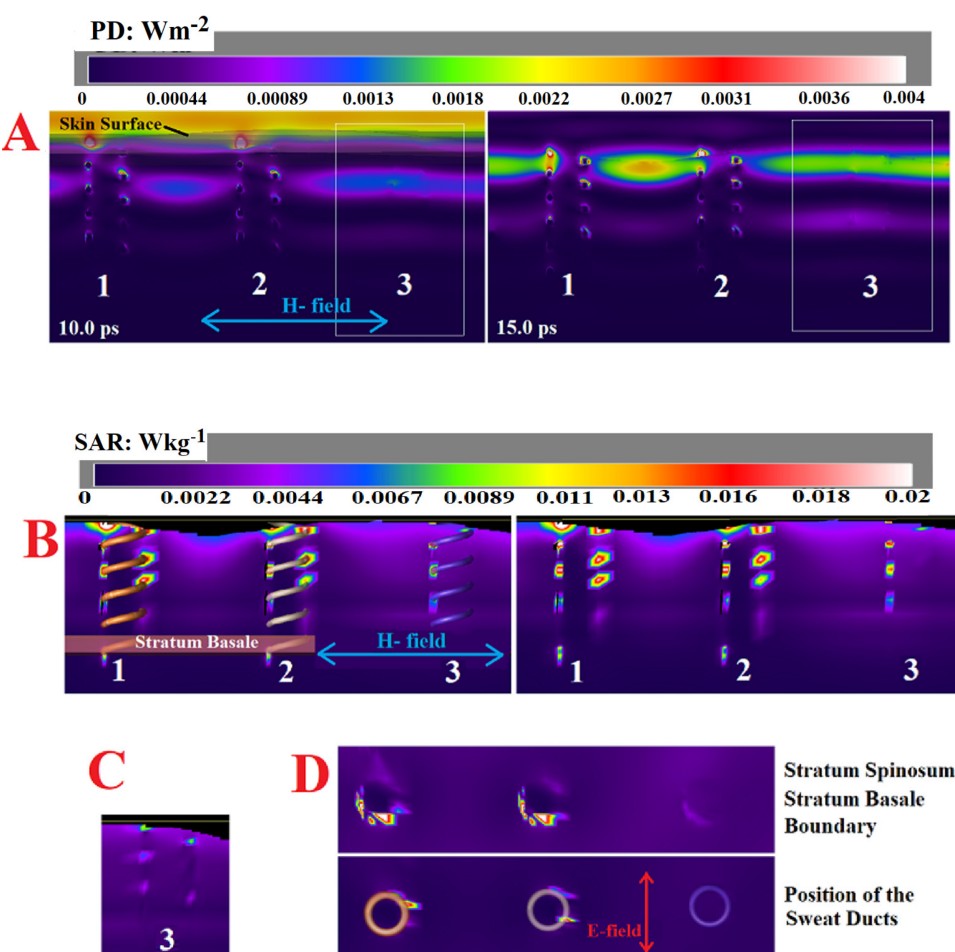

**Figure 6.** Simulations of PD and SAR surrounding sweat ducts in thick skin. The ducts (1) and (2) represent sweat ducts containing 10,014 Sm$^{-1}$ sweat, with an insulating layer surrounding the duct in (1). Duct (3) is filled with "physiological" sweat (105 Sm$^{-1}$). (**A**) Vertical section, direction of the polarized H-field as shown, PD at 10.0 and 15.0 ps into the simulation. There is little difference between the PD in the insulated and uninsulated ducts containing 10,014 Sm$^{-1}$ sweat. The duct containing the "physiological" sweat is barely discernable. (**B**) SAR, vertical cross-section, direction of the polarized H-field as shown. Left panel shows the general position of the sweat ducts and the Stratum Basale. The right edge of the duct containing the "physiological" sweat (105 Sm$^{-1}$) is barely discernable. (**C**) "Physiological" sweat containing duct, vertical section, the E-field as shown, polarization at 90° to the direction in (**B**). (**D**) Horizontal cross-sections at 0.0.35 mm depth (the level of the Stratum Spinosum/Stratum Basale boundary) and at 0.38 mm (within the Stratum Basale).

The SAR in both thick and thin skin exhibited an erratic pattern, broadly favoring the H-field direction of the incoming excitation. The sweat ducts containing the "high electrical conductivity sweat" of 1001 Sm$^{-1}$ and 10,014 Sm$^{-1}$ demonstrated PD and SAR shielding of the central region within the helix in thick skin. The presence of an insulating layer had very little effect on the SAR distribution. The polarization of the incident radiation affected the pattern of the reactive near field-induced SAR distribution. If the alignment of the E-filed was along the duct, the maximum SAR was above and below the sweat duct. With a tangential E-field alignment, the maximal SAR was on the lateral sides of the ducts. This was evident in both the helical and "unwound" ducts, but it was considerably easier to recognize in the "unwound" versions (Figure 7).

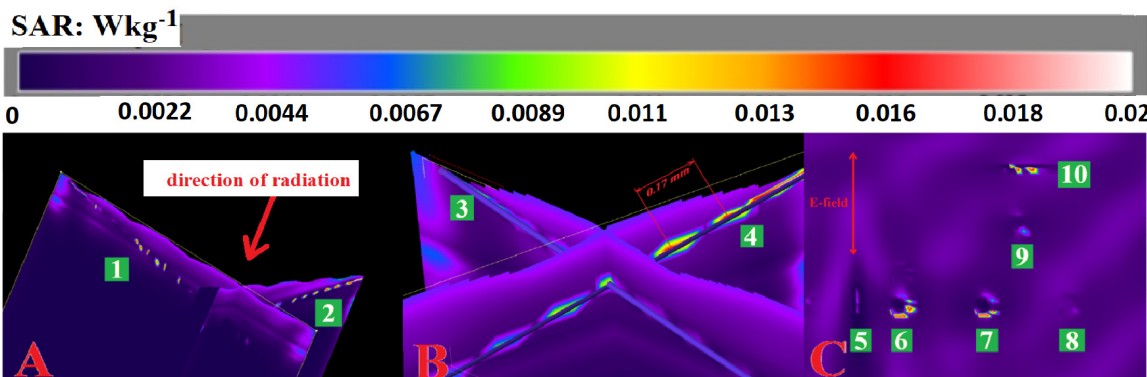

**Figure 7.** Simulations of SAR surrounding "unwound" sweat ducts in thick skin. (**A**) Unwound duct, with E-field polarization shown. (**B**) Detail of unwound sweat ducts. The SAR pattern suggests that there are standing waves within and surrounding the duct (1–4). The waves of ~0.17 mm (~$\lambda/2$) are situated within, above and below the duct when the duct is running parallel to the E-field direction and are lateral to the duct when the duct is perpendicular to the incident E-field. (**C**) Horizontal cutplane detail of ducts in the middle of the Stratum Corneum, E-field direction shown, (5) unwound duct, parallel, (6) helical sweat duct containing 10,014 Sm$^{-1}$ sweat, insulating layer surrounding duct, (7) no insulating layer, (8) helical sweat duct "physiological" sweat, (9) helical sweat duct 1001 Sm$^{-1}$ sweat, (10) unwound duct, perpendicular.

The SAR pattern surrounding the "unwound" sweat ducts suggests that there are waves of ~0.17 mm within the sweat with evanescent fields being generated within the surrounding tissues. which is approximately ~$\lambda/2$ of the 0.45 THz incident excitation within the tissues. The pattern was evident above and under the unwound duct when the duct was running parallel to the polarization, and it was lateral to the duct when the duct was perpendicular to the incident E-field direction. The tissue penetration represented by the SAR pattern in the reactive near field of the helical sweat ducts was very similar to the "unwound" sweat duct pattern.

In the Stratum Spinosum, the maximum SAR surrounding the sweat ducts in thin skin with simulations using "physiological" sweat were 20 times the intensity of the peak surrounding the hair. The increase in the Stratum Basale was 40% of the level in the Stratum Spinosum. The "high electrical conductivity" sweat increased the maximal SAR by up to 150% of the "physiological" sweat in the SS and led to increased SAR levels in the Stratum Basale to 400% of the "physiological" sweat levels.

## 4. Discussion

Finite Difference Time Domain simulations of hair and sweat ducts was undertaken using a polarized far-field excitation of 0.45 THz. The simulations yielded high-resolution PD and SAR images.

Both the simple and the complex hair have a lower refractive index and absorption coefficient compared to the surrounding tissue. Hair simulations produced a pattern in which the E-field and the SAR were enhanced in the direction of the magnetic field (H-field) and reduced in the direction of the E-field. This was a feature of all hair models (simple, complex and long).

The greatest H-field direction enhancement was noted at a depth of 0.05 mm, which corresponds to the upper Stratum Spinosum. The enhancement was 500% above the background values in the surrounding tissue at the stated depth of 0.05 mm. The pattern continued to a depth of 0.15 mm, where the enhancement was 100–150%. The regions of increased SAR and PD were approximately 0.1 mm in diameter (the diameter of the hair shaft), which would translate to 3 to 10 cells' thickness.

In contrast to the hair, which presents a uniform profile, a helical sweat duct presents a varying profile to the incoming radiation. The profile changes continuously, from the

duct bisecting the E-field to the duct running parallel to the E-field with each quarter turn. As with the hair shaft, the duct also has a varying angle to the incoming incident radiation. The angle changes along the width of the duct, from 12.5° to eventually reaching 90° at the sides of the duct. Both the "physiological sweat" and the "high electrical conductivity sweat" configurations have a higher refractive index and absorption coefficient within the duct, which is an inverse of the situation regarding the hair.

This results in all the radiation being admitted into the sweat being a traveling wave but varying amounts of total internal reflection within the sweat, resulting in evanescent wave formation in the region just outside the sweat duct.

In thin skin, the tissue SAR in regions of the Stratum Basale next to the sweat ducts was enhanced with all sweat conductivities in thin skin, including the ducts containing physiological sweat. In thick skin, the increase in tissue SAR in regions of the Stratum Basale is entirely dependent on the assumption of "high electrical conductivity sweat" as put forward by Feldman et al. [19]. The only effect of the helical structure appeared to be a reduction in SAR in the inner region of the duct when the duct was simulated with "high conductivity" sweat. In all other respects, the helical ducts behaved like the unwound, straight ducts. The tissue SAR near the anatomically correct helical and the "unwound" sweat ducts was very similar. This is the case for both thick and thin skin. The SAR pattern in both thick and thin skin was erratic, broadly favoring the direction of the H-field. The simulations suggest that any effect is only due to the conductivity of the sweat contained in the ducts and not any helical structure.

The dimensions of the regions of increased SAR due to the reactive near-field effects with the hair was in the order of 0.05–0.1 mm. The penetration represented by the SAR pattern in the reactive near field near hair models shows only a slight increase above the background below the level of the Stratum Spinosum, and thus, it is unlikely to lead to any adverse non-thermal effects given that the cells in the Stratum Spinosum are undergoing programmed cell death.

## 5. Conclusions

Given the high absorption coefficient of water at THz wavelengths, the effective penetration of THz into water-dominated biological substances is limited to a few tenths of a millimeter. For example, less than 5% of the incident 0.45 THz radiation survives 0.3 mm in tissues with the hydration level of the Stratum Spinosum. Even in the dry, dead Stratum Corneum, at 15% hydration, 5% of the incident 0.45 THz radiation survives only to about 0.55 mm. The result is that from purely analytical calculations, it is very difficult to create a significant radiative far field at THz frequencies by any antenna embedded in the skin. The addition of high-conductivity sweat does not overcome the opaque nature of the surrounding tissues at THz, and it does not produce evidence of a radiative far field. The addition of high-conductivity sweat does produce regions of greater SAR when compared to the physiological seat, but given the size of the enhanced SAR regions, there would be no general tissue-wide impacts.

The simulation results for anatomically correct helical and the "unwound" sweat ducts were very similar, suggesting that any enhancement is not due to the helical nature of the ducts but rather due to the assumptions regarding the conductive nature of the sweat. Any claims of increased tissue penetration thus rest on the claims based on mechanisms that elevate the conductivity of sweat above the physiological sweat levels.

When the exposures are adjusted for the ICNIRP (2013) [27] guidelines, and given the size of the enhanced SAR regions, any exposure at the maximum recommended power density of 1 kWm$^{-2}$ is unlikely to have a significant thermal effect. Exposures at higher than recommended levels, however, may compromise individual cells within the Stratum Basale. Given the lack of information regarding non-thermal effects on the Stratum Basale, it is not possible to speculate regarding any non-thermal changes within this layer when the skin is exposed to 0.45 THz radiation at the maximum PD outlined in the ICNIRP (2013) [27] guidelines. Such questions may need to be answered with long-term THz exposure studies.

**Author Contributions:** Conceptualization, Z.V. and A.W.W.; methodology, Z.V.; validation, Z.V. and A.W.W. writing—original draft preparation, Z.V.; writing—review and editing, N.F. and A.W.W.; supervision, A.L. and A.W.W.; project administration, A.W.W.; funding acquisition, A.W.W. All authors have read and agreed to the published version of the manuscript.

**Funding:** National Health and Medical Research Council of Australia (Project: APP1135076).

**Data Availability Statement:** Data is available on request from authors.

**Conflicts of Interest:** The authors declare no conflict of interest.

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
