# Peer review of "FDTD Simulations of Sweat Ducts and Hair at 0.45 THz"

_dermato, doi:10.3390/dermato3010006_

Round 1

Reviewer 1 Report

Manuscript Number:  Dermato-2217166 (MDPI)

Article Title:  FDTD Simulations of Sweat Ducts and Hair at 0.45 THz

It is very interesting to perform Terahertz irradiation of skin and skin hair to study various skin structures. Although the authors discuss about FDTD simulation to the skin, just simulation is not practical to apply human skin structure. It is useful to discuss about the detail application to actual human skin or chicken skin or so.

Comments:

Line 14 on page 1, in the abstract, what is the SAR?

Line 71 on page 2. Authors mentioned “The helical antenna equation”. What is the equation?

Line 91 on page 2. Authors mentioned “SS”. What is the SS?

Line 171 on page 4. Authors mentioned “32 papers” but authors cite only one reference. Why? 

On page 5 and 9. The table 1abd 2 need to add dividing lines. Time steps (units)

On page 16 in the discussion section, authors need to discuss more details about thermal effect on the skin because the THz absorption in the skin is significant.

Minor points:

General comments about manuscript: one paragraph composed of two sentences is not recommended for non-English country Journal readers. In addition, the manuscript needs to write more carefully.

Line 8 on page 1, in the abstract, Terahertz (THz) radiation?

Line 13 on page 1, in the abstract, what is the SAR?

Line 14 on page 1, in the abstract, Finite Difference Time Domain (FDTD)?

Line 15 on page 1, mm, respectively?

And many typing mistakes about periods.

Author Response

Reply to author #1

Thank you for you review. Your remarks have helped us to improve the paper.  We hope we have addressed your remarks in a satisfactory manner.

It is very interesting to perform Terahertz irradiation of skin and skin hair to study various skin structures. Although the authors discuss about FDTD simulation to the skin, just simulation is not practical to apply human skin structure. It is useful to discuss about the detail application to actual human skin or chicken skin or so.

 Comments:

Line 14 on page 1, in the abstract, what is the SAR?

SAR “is the specific absorption rate”, the rate at which energy is absorbed, rather than just passing through a material.

We have added the words “specific absorption rate” to the abstract.

Line 71 on page 2. Authors mentioned “The helical antenna equation”. What is the equation?

The equation if found in lines #71-73, i.e. optimal response is at  3λ/4 ≤ 2πr ≤ 4λ/3, where λ is the wavelength in the medium that surrounds the antenna and r is the radius of the helix. 

Line 91 on page 2. Authors mentioned “SS”. What is the SS?

 SS is the acronym for the Stratum Spinosum

We have added the  (SS) at the first mention of the Stratum Spinosum in line # 65,

Line 171 on page 4. Authors mentioned “32 papers” but authors cite only one reference. Why? 

The reference contains the analysis of the 32 papers. We have altered the text to make this clear.

 On page 5 and 9. The table 1 and 2 need to add dividing lines. Time steps (units)

 Timesteps is simply the number of timesteps and does not have units, we have altered the line to “Number of Timesteps” to clarify the meaning. We have added extra dividing lines to help with the visual scanning of the tables.

On page 16 in the discussion section, authors need to discuss more details about thermal effect on the skin because the THz absorption in the skin is significant.

The thermal effects can, indeed, be significant if the incoming source is sufficiently high.

We have decided present this as a separate paper, since the  diffusion characteristics of the skin components are complex and deserve an extensive analysis.

The deeper skin structures reliant on the Pennes bioheat transfer equation and the outer skin components on radiative transfer that is dependent on ambient temperature, clothing and evaporative heat transfer.  

We are in the process of presenting a paper on this subject.

 Minor points:

General comments about manuscript: one paragraph composed of two sentences is not recommended for non-English country Journal readers. In addition, the manuscript needs to write more carefully.

Line 8 on page 1, in the abstract, Terahertz (THz) radiation?

We have expanded the line to read “ Terahertz frequency electromagnetic radiation (THz)”

Line 13 on page 1, in the abstract, what is the SAR?

See above.  

Line 14 on page 1, in the abstract, Finite Difference Time Domain (FDTD)?

 We have added (FDTD) after  “Finite Difference Time Domain” in the abstract.  

Line 15 on page 1, mm, respectively?

corrected

And many typing mistakes about periods.

We have corrected the errors we could find.  Once the paper is approved, we will submit final draft to a professional proofreader.

Reviewer 2 Report

The authors present anatomically matched FDTD simulations of hair and sweat ducts using a Terahertz excitation source. Overall, the manuscript is well-written, correctly formatted, and is of interest to the research community—a few cursory changes, as suggested below to improve readability.

General Comments:

1. Results and Discussions must be separated. A lot of commentary currently resides in results and else where, which will be better suited to be placed in the Discussions.

2. Add a Conclusion section. Currently non-existent in the present manuscript.

3. Authors must highlight significant takeaways of the work in the abstract.

4. Add a clinical relevance section, which will situate the work in the context of a biomedical research problem.

5. Authors must report a metric to be able to visualize reconstruction errors.

Specific Comments:

1. Why was a mono-excitation source used instead of a broadband ultrashort Terahertz source, if the intent is to somehow simulate ThZ interactions on dermal tissues?

Summarily, the manuscript will benefit from a minor revision, with the above points considered as a guide to improve readability.

Author Response

Reply to author #2

Thank you for you review. Your remarks have certainly helped us to improve the paper. We have added the conclusion as suggested, and made other adjustments. We hope we have addressed your remarks in a satisfactory manner.

Conclusion:

Given the high absorption coefficient of water at THz wavelengths, the effective penetration of THz into water dominated biological substances is limited to a few tenths of a millimeter. For example, less than 5% of the incident 0.45 THz radiation survives 0.3 mm in tissues with the hydration level of the Stratum Spinosum. Even in the dry, dead, Stratum Corneum, at 15% hydration, 5% of the incident 0.45 THz radiation survives only to about 0.55 mm. The result is that, from purely analytical calculations, it is very difficult to create a significant radiative far field at THz frequencies by any antenna embedded in the skin. The addition of high conductivity sweat does not overcome the opaque nature of the surround tissues at THz, and does not produce evidence of a radiative far field . The addition of high conductivity sweat does produce regions of greater SAR when compared to the physiological seat, but given the size of the enhanced SAR regions, there would be no general tissue wide impacts.   

The simulation results for anatomically correct helical and the “unwound” sweat ducts were very similar, suggesting that any enhancement is not due to the helical nature of the ducts, but due to the assumptions regarding the conductive nature of the sweat. Any claims of increased tissue penetration thus rest on the claims based on mechanisms that elevate the conductivity of sweat above the physiological sweat levels.

When the exposures are adjusted for the ICNIRP (2013) [27] guidelines, and given the size of the enhanced SAR regions, any exposure at the maximum recommended power density (PD) of 1kWm-2 is unlikely to have a significant thermal effect. Exposures at higher than recommended levels, however, may compromise individual cells within the Stratum Basale. Given the lack of information regarding non-thermal effects on the Stratum Basale, it is not possible to speculate regarding any non-thermal changes within this layer when the skin is exposed to 0.45 THz radiation at the maximum PD outlined in the ICNIRP (2013) [27] guidelines. Such questions may need to be answered with long term THz exposure studies.

The authors present anatomically matched FDTD simulations of hair and sweat ducts using a Terahertz excitation source. Overall, the manuscript is well-written, correctly formatted, and is of interest to the research community—a few cursory changes, as suggested below to improve readability.

General Comments:

  1. Results and Discussions must be separated. A lot of commentary currently resides in results and else where, which will be better suited to be placed in the Discussions.

We have changed the manuscript as suggested.

  1. Add a Conclusion section. Currently non-existent in the present manuscript.

We have added a conclusion section as suggested.

  1. Authors must highlight significant takeaways of the work in the abstract.

We have added this in the conclusion section as suggested.

  1. Add a clinical relevance section, which will situate the work in the context of a biomedical research problem.

We have added this in the conclusion section as suggested.

  1. Authors must report a metric to be able to visualize reconstruction errors.

We have added a metric for this as a section in the methods.

Specific Comments:

  1. Why was a mono-excitation source used instead of a broadband ultrashort Terahertz source, if the intent is to somehow simulate THz interactions on dermal tissues?

We have opted for a single frequency because the dielectric parameters of water, and hence biological substances, change in a significant, non-linear way in the THz range, thus setting broadband simulation parameters is not going top produce reliable results.

Summarily, the manuscript will benefit from a minor revision, with the above points considered as a guide to improve readability.

Round 2

Reviewer 1 Report

Acceptable